# Surgical Outcomes and Complications of Custom-Made Prostheses in Upper Limb Oncological Reconstruction: A Systematic Review

**DOI:** 10.3390/jfmk9020072

**Published:** 2024-04-11

**Authors:** Camillo Fulchignoni, Silvia Pietramala, Ivo Lopez, Giovan Giuseppe Mazzella, Chiara Comisi, Carlo Perisano, Lorenzo Rocchi, Tommaso Greco

**Affiliations:** Hand Surgery and Orthopedics Unit, Department of Orthopaedics and Traumatology, Catholic University of Rome, Fondazione Policlinico Universitario A. Gemelli IRCCS, 00168 Rome, Italy; camillo.fulchignoni@gmail.com (C.F.); silvia.pietramala01@gmail.com (S.P.); dott.ivolopez@gmail.com (I.L.); ggmazzella@gmail.com (G.G.M.); chiara.comisi22@gmail.com (C.C.); carlo.perisano@policlinicogemelli.it (C.P.); lorenzo.rocchi@policlinicogemelli.it (L.R.)

**Keywords:** upper-limb tumors, custom-made prosthesis, upper-limb oncological reconstruction, bone tumors, musculoskeletal oncology

## Abstract

Bone tumors of the upper limb are a common cause of bone pain and pathological fractures in both old and young populations. Surgical reconstruction and limb salvage have become valid options for these patients despite this kind of surgery being challenging due to the need for wide bone resection and the involvement of surrounding soft tissues. Computer-assisted technology helps the surgeon in pre-operative planning and in designing customized implants. The aim of this study was to investigate the surgical outcomes and complications of custom-made prostheses in oncologic reconstruction of the upper limb and if they are reliable options for patients suffering from aggressive tumors. An electronic search on PubMed, Google Scholar, and Web of Knowledge was conducted to identify all available articles on the use of custom-made prostheses in oncological resections of the upper limb. Twenty-one studies were included in the review, comprising a total of 145 patients with a mean age of 33.68 years. The bone involved was the humerus in 93 patients, and the radius was involved in 36 patients. There were only six cases involving proximal ulna, three cases involving the scapula, and seven cases involving the elbow as well as soft tissues around it. The most frequent primary tumor was the giant cell tumor, with 36 cases, followed by osteosarcoma with 25 cases, Ewing Sarcoma with 17 cases, and Chondrosarcoma with 7 total cases. Forty patients were affected by bone metastases (such as renal cell cancer, breast cancer, melanoma, and rectal cancer) or hematologic diseases involving bone (lymphoma, myeloma, or non-Hodgkin disease). Custom-made prostheses are a viable option for patients who suffer from malignant tumors in their upper limbs. They are a reliable aid for surgeons in cases of extensive resections.

## 1. Introduction

Bone tumors in the upper limbs are a common cause of bone pain and pathological fractures and can lead to a decreased quality of life [1,2] The recommended treatment for this condition is tumor resection followed by reconstruction, with the aim of achieving stability and pain control [3,4]. However, surgical reconstruction can be challenging due to intrinsic factors such as tumor location and grade as well as the morpho-anatomical structure of each individual patient. Therefore, a customized approach is always necessary to evaluate and study each case effectively. Computer-assisted technology is playing an increasingly important role in surgery for musculoskeletal tumors, including pre-operative planning, intraoperative computer navigation for accurate identification of the extent of the tumor, and precise fitting of custom-made prostheses [5,6,7]. Computer-aided design and computer-aided manufacturing (CAD-CAM) offer a new approach to designing custom-made implants [8,9]. Rapid prototyping (RP) is a set of CAD-CAM technologies used to rapidly create a solid object of almost any shape from three-dimensional image data. It is usually achieved through the additive manufacturing technology of 3D printing. RP techniques first became available in the late 1980s and are now used in a wide range of applications [10,11]. RP technology has been utilized to create tailored bone implants designed to match the size and shape of the patient’s bone defect. In this process, the unaffected side of the bone is used as a reference to create a mirror image data source, which is then used to manufacture the implant. This ensures that the implant is customized to fit the patient’s unique needs and specifications [12].

Three-dimensional printers that are commercially available come with software that can convert CT and MRI data into 3D models that can be easily printed. This has led to more widespread use in the medical field, especially in upper extremity surgery, where anatomical abnormalities can be better appreciated by surgeons during operations. While some studies have reported no significant advantages using 3D printing compared to conventional approaches [12], there has been a growing trend in the medical application of 3D printing technology in recent years, particularly for upper extremity surgery. This can be attributed to the fact that prices have become much more affordable compared to the past. This development is a significant step forward in the treatment of oncological pathologies of the upper limb, such as malignant chondromas, for which surgery is the gold standard due to its poor susceptibility to chemotherapy and radiotherapy [13]. In addition, limb-salvage surgery offers better functional outcomes and quality of life without a reduction in survival or an increase in morbidity when compared to amputation. This is particularly important in cases where intra-compartmental tumors require resection with a wide surgical margin of healthy tissues, or in high-grade tumors that infiltrate vessels and nerves or cause pathological fractures; these cases were systematically treated with amputation up until a few years ago [14].

The aim of this study was to investigate the use of custom-made prostheses in oncologic reconstruction of the upper limb based on surgical outcomes and possible complications.

## 2. Materials and Methods

A literature search was performed in agreement with the Preferred Reporting Items for Systematic Reviews and Meta-Analyses (PRISMA) guidelines [15] (PRISMA checklist in Appendix A).

### 2.1. Search Strategy

A literature search was performed on MEDLINE through PubMed, Google Scholar, and Web of Knowledge. Searches were performed on September 2023 using the following terms: “custom-made upper limb OR shoulder OR humerus OR elbow OR forearm OR wrist OR radius AND tumor OR oncologic”. No restrictions in terms of data of publication were applied to the research. To avoid missing studies, no filters were applied to the search strategies. Level I-V evidence studies, according to the American Academy of Orthopedic Surgeons [16], were included in the initial research, so that the bibliography of reviews that were ultimately not included in the study could also be analyzed in search of any further studies that met the inclusion criteria.

### 2.2. Study Selection

The inclusion criteria were (1) information about the bone segment or joint involved in the tumor and subjected to surgery; (2) description of surgical techniques and type of prosthesis used; (3) surgical and clinical outcomes; (4) information about complications; (5) possibility to sort data by the segment treated (in the case of studies including both upper and lower libs or different upper-limb segments).

The exclusion criteria included (1) any language not known by the authors (English, Italian, and French); (2) full text not available; (3) review article; (4) insufficient data regarding the pathology.

It was decided to also include case reports because of the limited literature on the subject, to have a wider range of case studies available.

Using titles and abstracts, three authors independently selected studies for inclusion (S.P., G.G.M. and I.L.). Any discordances were solved by consensus with a senior author (C.F.).

All abstracts were reviewed to determine adherence to the inclusion and exclusion criteria of our study. If no abstract was published or if the abstract did not have sufficient information to determine eligibility, the full-length manuscript was reviewed. Articles with questionable data were discussed with the senior author.

### 2.3. Data Extraction and Analysis

Three observers (S.P., G.G.M. and I.L.) independently searched and collected data from the included studies in an Excel worksheet (Microsoft Corporation, Redmond, WA, USA. Microsoft Excel [Internet]. 2018. Available from: https://office.microsoft.com/excel accessed on 9 April 2024). The following data were collected by included studies: number of patients involved, demographic features, bone segment or joint, type of tumor, necessity of chemo- or radiotherapy, type of prosthesis used, resection margins, necessity of flap or bone substitution, post-surgical complications, mean follow-up, and Musculoskeletal Tumor (MST) score [17] when available. The methodological quality of the studies was assessed using the modified Coleman Methodology Score (mCMS) [18]. Two independent investigators evaluated each article (A.D.F. and M.B.B.); in cases with more than a five-point difference between their ratings, the discrepancy was solved by consensus with a third author (R.V.). The mCMS ranges from 0 to 100 points, classifying the included studies based on the final score as excellent (85–100 points), good (70–84 points), fair (50–69 points), or poor (<50 points).

Statistical analysis was performed using SPSS 18.0 for Windows (SPSS Inc., Chicago, IL, USA). Descriptive statistics were used to summarize the findings across all the included studies.

## 3. Results

### 3.1. Search and Literature Selection

The review examined studies that used custom-made prostheses in upper limb reconstruction for patients with malignant tumors, including primary or metastatic bone and soft tissue tumors. The electronic search resulted in 86 articles. After exclusion of duplicates, 63 studies remained; titles and abstracts were screened, resulting in 26 remaining articles. After full-text screening, five articles were excluded. Thus, 21 studies meeting the inclusion criteria were finally included in this review, following the PRISMA flow chart [8,19,20,21,22,23,24,25,26,27,28,29,30,31,32,33,34,35,36,37,38] (Figure 1).

### 3.2. Characteristics and Evaluation of Studies

Of the 21 studies, 14 (66.66%) were retrospective, 6 (28.57%) were case reports, and 1 (4.77%) was a prospective study. According to the evaluation with the mCMS, the rating of the studies was poor in 20 studies and fair only in one study, with a value of 41.28 (maximum 53—minimum 31), thus with an overall poor level.

### 3.3. Demographics

In articles regarding both upper and lower limbs, data were extracted and sorted by the different bone districts [19,20,21] (Table 1).

The studies counted 145 patients overall, 59.31% of patients (86) were male and 40.69% (59) were female. The youngest patient was 5 years old, and the oldest was 87 years old, with a mean age of 33.68 years old. The mean follow-up was 43.37 months.

### 3.4. Bone Segment or Joint Involved

The bone involved was the humerus in 93 patients (66.66%), divided into the proximal humerus in 35 patients (24.13%), distal humerus in 37 patients (25.51%), and diaphyseal humerus in 21 patients (14.48%).

The radius was involved in 36 patients (24.82%): the proximal radius was involved in two cases only (1.37%), while most cases (34) involved the distal radius (23.44%). There were only six cases (4.13%) involving the proximal ulna, while there were three cases (2.06%) involving the scapula and seven cases (4.82%) involving the elbow as well as the soft tissues around it.

### 3.5. Type of Tumor

The most represented primary tumor was the giant cell tumor, with 36 cases (24.82%), followed by osteosarcoma with 25 cases (17.24%). The third-most represented tumor was Ewing Sarcoma, found in 17 patients (11.72%). Chondrosarcoma was the primary diagnosis in seven total cases (4.82%). Forty patients (27.58%) were affected by bone metastases (such as renal cell cancer, breast cancer, melanoma, and rectal cancer) or hematologic diseases involving bone (lymphoma, myeloma, or non-Hodgkin disease).

The remaining patients were affected by various conditions, such as rhabdomyosarcoma (0.68%), synovial sarcoma (1.28%), epithelioid hemangioendothelioma (0.68%), pleomorphic sarcoma (0.68%), aggressive fibromatosis (0.68%), malignant fibrous histiocytoma (1.92%), soft tissue sarcoma (0.68%), and giant cell sarcoma (0.64%) (Table 2).

### 3.6. Other Treatments

In all cases, neoadjuvant or adjuvant therapy was administered according to the prescription of an oncologist, based on the type and stage of the tumor.

All patients were evaluated pre-operatively by X-ray, CT-scan, MRI and needle biopsy to confirm the diagnosis and to obtain data for the custom-made implant which, in the majority of cases, was designed using three-dimensional (3D) technology.

It was impossible to extract data regarding the quality of margins at the resection site, but most authors stated that they always tried to achieve wide-margin resections when possible.

Only four authors [20,21,22,31] specified that they needed a flap to reconstruct soft tissue: two authors used a latissimus dorsi flap, one author a triceps flap, and in another case a myofascial flap of the latissimus dorsi, while five authors reported using bone grafts (both cancellous bone chips and allograft or autograft, such as a fibula graft).

### 3.7. Surgical Complications

As regards post-surgical complications, we did not consider the development of metastases as a complication but as an undesirable event related to the biology of the primary disease.

In 13 cases (8.96%), aseptic loosening of the implant was reported, and there were only five cases (3.44%) of reported infection, such as wound dehiscence or deep infection. The aseptic loosening was treated by Rolf and Gohlke [37] by a custom-made ulnar component with cementless fixation, 2 years after the primary prosthesis positioning. The same complication was observed by Tang et al. [21], but they did not specify the treatment used. On the other hand, Wang et al. [38], despite observing loosening, did not perform any additional surgery, as the patient showed no symptoms. Hanna et al. [8] reported aseptic loosening in three patients: two patients underwent a new implant insertion, while the third one had the humeral component recemented and the high-density polyethylene bushed replaced.

As regards infections, surgical revision was performed in the case of superficial wound dehiscence followed by antibiotic treatment [8,20], while in the case of deep infection, revision was eventually performed [8]. One case of late infection was recorded (50 months after primary surgery), caused by plate exposure and treated by debridement and plate removal [28].

In eight cases (5.51%), a complete or partial dislocation was reported; however, though the treatment of choice was not described, it was stated that satisfactory stability was achieved in all cases [19,28,32,38].

As intraoperative complications, vascular injuries were reported in only one case (0.68%) [21], while nerve injury (reported as palsy or neurapraxia) occurred in four cases (2.75%). Nerve injury was resolved spontaneously after three months in the case described by Pruksakorn et al. [32] and after six months in the case described by Hanna et al. [8]. McGrath et al. [35] reported sacrifice of the radial nerve due to tumor involvement, while the ulnar nerve neuropraxia resolved spontaneously. The vascular injury described by Tang et al. [21] was reported intra-operatively and immediately solved with preserved upper-limb function.

Hardware-related complications, such as protruding screws or implant exposure, were reported in three cases (2.06%). Szostakowski et al. [25] reported protruding screws on the dorsal aspect of the wrist, treated by removal of the screws in a day procedure without sacrificing implant functionality (wrist arthrodesis). Periprosthetic fractures were observed in four cases (2.75%). Hanna et al. [8] decided to treat peri-prosthetic humeral fracture by inserting a longer humeral stem, while McGrath et al. [35] revised with a distal humeral replacement in one case due to insufficient bone stock and with a custom-made distal component in another case.

Thirty-two deaths (20.51%) were reported, but these data cannot be considered reliable because follow-up periods were different and not all authors reported this type of data. Also, the type of tumor was different among the patients, so the meaning of such data is inconsistent.

### 3.8. Functional Score

MST score was calculated in 112 patients (77%), with a mean value of 59.35%. Among these, patients of five articles [20,23,26,35,36] had an MST score inferior to 50%. All such patients had high-grade osteosarcoma or bone metastases, and complications were reported. Excluding those five papers, the MST score calculated in the remaining 95 patients was 75.52%.

## 4. Discussion

Surgical procedures to remove tumors from the upper limb and reconstruct bone defects are becoming more common. Prior to the 1970s, amputation was the primary approach to treating malignant bone tumors such as osteosarcoma, Ewing sarcoma, and chondrosarcoma due to their aggressive nature [39]. While amputation was effective, it was a distressing procedure and had low 5-year survival rates, ranging from 10% to 20% [40]. Even with amputation, approximately 80% of patients still succumbed to metastatic disease. However, limb-salvage surgery has become the preferred treatment for primary bone tumors and bone metastasis, with a focus on restoring function for patients affected by these types of tumors [41]. Five-year survival rates have significantly improved to 66–82%, with notable improvements in function and patient-reported outcomes compared to amputation [40]. However, preserving limb function in aggressive tumors can be challenging due to the complexity of bone resection and the need for complex implants that alter the limb’s anatomy. The current options in terms of limb-salvage techniques involve osteoarticular allografts, bone autografts, prostheses, and graft prostheses [42].

Bone autografts can be a useful option in cases of wide resections, either alone or in combination with an implant such as an endoprosthesis. Autografts are a viable surgical option primarily due to their biological origin, which promotes hypertrophy in response to increased load on the limb and promotes the vascularity of tissues [43]. However, it cannot be used in the setting of post-operative radiation [44].

Bulk allografts are obtained from a donor and can be used to reconstruct significant bone and soft tissue defects, allowing for complex primary reconstructions that can support mechanical loads involving the joint [45]. However, they have some notable disadvantages, such as complications at the host–donor junction, due to the lack of vascular supply. Nonetheless, they continue to be a viable surgical option, mainly because the allograft progressively incorporates into the host tissue over time [43].

In case of extensive bone loss, megaprostheses offer the possibility to allow great bone resection and limb salvage; however, there are several concerns about complications in these large reconstructions [46].

Custom-made 3D-printed prostheses are becoming increasingly popular for limb salvage, leading to a significant improvement in patients’ quality of life [7]. According to Smith and Burgess [47], many prostheses are now created using CAD-CAM technology. This approach starts with simple measurements rather than exact anatomic data obtained through casting, scanning, or digitalization. Furthermore, it is also gaining more attention for pediatric patients [20].

The humerus is the fourth-most common site for primary bone tumors and the second-most common site for bone metastases [1,48]. This is consistent with the results obtained from our review, since the humerus was found to be involved for 93 patients (66.66%). Although the proximal humerus is the most frequent site involved according to the literature, in our study we found a slightly higher rate in the distal humerus (37 cases in the distal humerus vs. 35 cases in the proximal humerus).

Our findings indicated that the giant cell tumor (24.82%) was the most prevalent bone tumor, followed by osteosarcoma (17.24%). This might be because most of the population we studied consisted of young adults. According to the National Cancer Institute [49], chondrosarcoma is the most common primary bone tumor in adults, accounting for 40% of all cases, followed by osteosarcoma (28% of cases). However, in children and young adults, osteosarcoma is the most frequent primary bone tumor.

The most common malignancy reported overall was bone metastasis (27.58%), in line with what we found in the literature [50].

Limb-salvage surgery is a complex procedure, with a steep learning curve. However, it has been found to be a safe option with a low rate of complications. The use of custom-made products has made the procedure less challenging by allowing for guided surgery. According to the literature [8,51], the most common complications include aseptic loosening, infection, and nerve injury. Nonetheless, our review showed that these complications often resolve spontaneously in the case of nerve injuries and can be easily resolved through revision surgery in other cases.

Soft tissue coverage is a crucial issue in oncologic surgery of the upper limb [52]. Surprisingly, in the present review, only four authors used vascularized flaps to address soft tissue defects. The authors of this review believe that this can be explained by the recent emergence of orthoplastic surgery [53] and the frequent use of alternative techniques such as Negative Pressure Wound Therapy in the past [54].

### Limitations and Strengths of the Review

It is important to note the limitations of this review. Firstly, the decision to include case reports in the review due to the limited literature available as well as the small sample size of the included studies and their low methodological quality as evidenced by the mean mCMS are limitations of this review. Furthermore, the wide heterogeneity of the studies only allowed for a descriptive statistical analysis; heterogeneity was mainly due to the inclusion of different joints (given the small number of each one). Additionally, not all authors reported results using the MST score, which makes the final assessment non-uniform. Finally, although the economic aspects of using a custom-made prosthesis are relevant, this topic was not covered in this review.

The strength of the review is that it represents a first overview of the use of custom-made prostheses for the upper limb in oncological patients and may provide stimulus for further research and studies of higher methodological quality.

## 5. Conclusions

Custom-made prostheses are an option for patients who suffer from malignant tumors in their upper limbs. They are a reliable aid for surgeons in cases of extensive resections. Although amputations and arthrodesis are still considered reliable and easily achievable techniques, the use of custom-made prostheses, although surgically more challenging, makes it possible to restore the patient’s functional independence, especially for young patients, who are the most represented in our review.

## Figures and Tables

**Figure 1 jfmk-09-00072-f001:**
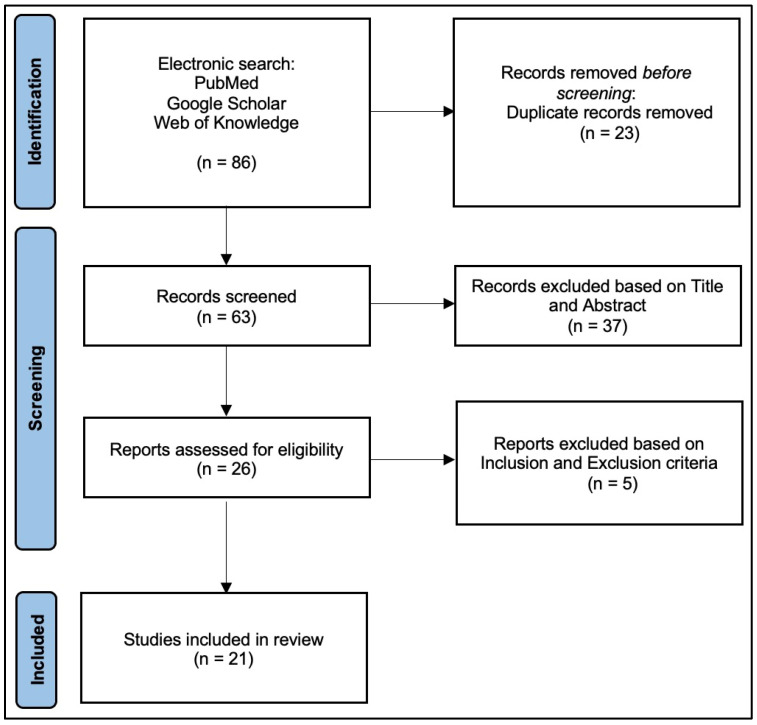
PRISMA2020 Flowchart.

**Table 1 jfmk-09-00072-t001:** Studies included in the review and main features.

Author, Year	Type of Study	No. Patients	M	F	Age	Location	Diagnosis	mCMS
Li et al., 2023 [19]	RE	7	5	2	29.3	Humerus/Radius	OS/ES	47
Vitiello et al., 2023 [22]	RE	1	0	1	80	Humerus	CS	31
Liang et al., 2022 [23]	CR	1	0	1	49	Humerus	Breast	37
Wang et al., 2022 [24]	CR	1	1	0	14	Radius	EH	42
Beltrami et al., 2021 [20]	RE	4	1	3	13	Humerus/Radius/Scapula	OS/EW/RS	45
Szostakowski et al., 2021 [25]	RE	4	2	2	42	Radius	GCT	43
Beltrami et al., 2021 [26]	CR	1	0	1	13	Humerus	OS	42
Kuptniratsaikul et al., 2021 [27]	CR	1	0	1	34	Radius	GCT	42
Wang et al., 2020 [28]	RE	15	6	9	38	Radius	GCT	38
Damert et al., 2020 [29]	CR	1	1	0	36	Radius	GCT	37
Hu et al., 2019 [30]	RE	7	3	4	34.9	Humerus	OS/CS/GCT	43
Beltrami et al., 2018 [31]	RE	2	0	2	26.5	Scapula	ES/SS	43
Wang et al., 2016 [38]	RE	10	7	3	39	Distal Radius	GCT	46
Pruksakorn et al., 2015 [32]	PR	16	/	/	37.5	Humers/Ulna	Metastasis	53
Damert et al., 2013 [33]	CR	1	1	0	36	Radius	GCT	46
Natarjan et al., 2012 [34]	RE	11	6	5	17	Humerus	OS/EW/AF	47
McGrath et al., 2011 [35]	RE	13	9	4	35	Humerus	ES/OS/CS/MFH/PS/Metastasis	35
Tang et al., 2009 [21]	RE	26	10	6	38.8	Humerus Ulna/Elbow	EW/OS/MFH/SS/Metastasis	37
Hanna et al., 2007 [8]	RE	18	11	7	48.2	Humerus	EW/GCT/OS/CS/Metastasis	35
Ahlmann and Menendez, 2006 [36]	RE	1	1	0	66	Humerus	Metastasis	43
Rolf and Gohlke, 2004 [37]	RE	4	4	0	32	Elbow	Metastasis	35

AF: aggressive fibromatosis; CR: case report; CS: chondrosarcoma; EW: Ewing’s Sarcoma; GCT: giant cell tumor; mCMS: modified Coleman Methodology Score; MFH: malignant fibrous histiocytoma; OS: osteosarcoma; PR: prospective; PS: pleomorphic sarcoma; RE: retrospective; RS: rhabdomyosarcoma; SS: synovial sarcoma.

**Table 2 jfmk-09-00072-t002:** Location and diagnosis of the tumor.

		Total Cases	Male	Female
		145	86 (59.31%)	59 (40.69%)
**Mean age**		33.68 years old		
**Follow-up**	Mean	43.37 months		
**MSTS score**		112 (77%)	59.35%	
**Location**	Proximal Humerus	35 (24.13%)	19	16
	Distal Humerus	37 (25.51%)	24	13
	Humeral Diaphysis	21 (14.48%)	13	8
	Proximal Ulna	6 (4.13%)	4	2
	Proximal Radius	2 (1.37%)	1	1
	Distal Radius	34 (23.44%)	18	16
	Scapula	3 (2.06%)	0	3
	Elbow	7 (4.82%)	7	0
**Diagnosis**	Giant Cells Tumor	36 (24.82%)	20	16
	Osteosarcoma	25 (17.24%)	18	7
	Ewing’s Sarcoma	17 (11.72%)	10	7
	Chondrosarcoma	7 (4.82%)	2	5
	Metastasis	40 (27.58%)	25	15
	Others	20 (13.79%)	11	9
**Complications**	Aseptic loosening	13 (8.96%)		
	Dislocation	8 (5.51%)		
	Infection	5 (3.44%)		
	Nerve Injury	4 (2.75%)		
	Periprosthetic fractures	4 (2.75%)		
	Hardware failure	3 (2.06%)		
	Vascular Injury	1 (0.68%)		

MSTS: Musculoskeletal Tumor Society Score.

## Data Availability

The data are available on request to the corresponding author.

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
