# Peer review of "Surgical Outcomes and Complications of Custom-Made Prostheses in Upper Limb Oncological Reconstruction: A Systematic Review"

_jfmk, 2024, doi:10.3390/jfmk9020072_

Round 1
Reviewer 1 Report
Comments and Suggestions for Authors
The manuscript by C. Fulcignoni et al. "The Use of Customized Prostheses in Oncologic Upper Extremity Reconstruction: A Systematic Review," is a critical evaluation of the available data in the literature on the use of artificial prostheses for upper extremity reconstruction in the surgical treatment of cancer.
For obvious reasons, prosthetic limb replacement after bone tumor removal is a highly desirable alternative to limb amputation. Customized prosthetics allow the replacement of the limb morphology by introducing artificial, allogeneic or autologous implant material into the postoperative area. The literature search conducted by the authors is based on identifying recent (by September 2023) publications dealing with upper limb, shoulder, humerus, elbow, elbow, forearm, wrist, and radius reconstruction. Case descriptions of instrumental surgery for osteosarcoma, Ewing's Sarcoma, chondrosarcoma, rhabdomyosarcoma, giant cell tumor, synovial sarcoma, aggressive fibromatosis, malignant fibrous histiocytoma, pleomorphic sarcoma, and metastasis formation are reviewed.
Among the operated patients, the most frequent diagnoses were giant cell tumor, osteosarcoma, Ewing Sarcoma, and Chondrosarcoma. A significant group had bone metastases of various origins. In most of the cases considered by the authors, the creation of customized implants was designed using 3-Dimensional technology.
The authors noted vascular injuries and nerve injury as intraoperative complications. Loosening of the implant, as well as hardware related complications were rarely observed and were corrected by additional postoperative manipulations.
Overall, the manuscript makes a favorable impression with the quality of the material presented.
The reviewer has no major comments on the work.
A small remarks on Table 2:
1. For a more complete perception of the information presented, the authors could be recommended to divide it into three columns (total-males-females), organizing the available information accordingly.
2. The table title is unfortunate and does not reflect the information contained. It is desirable to include the words "location" "diagnosis" in the table title

Author Response
The manuscript by C. Fulcignoni et al. "The Use of Customized Prostheses in Oncologic Upper Extremity Reconstruction: A Systematic Review," is a critical evaluation of the available data in the literature on the use of artificial prostheses for upper extremity reconstruction in the surgical treatment of cancer.
For obvious reasons, prosthetic limb replacement after bone tumor removal is a highly desirable alternative to limb amputation. Customized prosthetics allow the replacement of the limb morphology by introducing artificial, allogeneic or autologous implant material into the postoperative area. The literature search conducted by the authors is based on identifying recent (by September 2023) publications dealing with upper limb, shoulder, humerus, elbow, elbow, forearm, wrist, and radius reconstruction. Case descriptions of instrumental surgery for osteosarcoma, Ewing's Sarcoma, chondrosarcoma, rhabdomyosarcoma, giant cell tumor, synovial sarcoma, aggressive fibromatosis, malignant fibrous histiocytoma, pleomorphic sarcoma, and metastasis formation are reviewed.
Among the operated patients, the most frequent diagnoses were giant cell tumor, osteosarcoma, Ewing Sarcoma, and Chondrosarcoma. A significant group had bone metastases of various origins. In most of the cases considered by the authors, the creation of customized implants was designed using 3-Dimensional technology.
The authors noted vascular injuries and nerve injury as intraoperative complications. Loosening of the implant, as well as hardware related complications were rarely observed and were corrected by additional postoperative manipulations.
Overall, the manuscript makes a favorable impression with the quality of the material presented.
The reviewer has no major comments on the work.
A small remarks on Table 2:
- For a more complete perception of the information presented, the authors could be recommended to divide it into three columns (total-males-females), organizing the available information accordingly.
- The table title is unfortunate and does not reflect the information contained. It is desirable to include the words "location" "diagnosis" in the table title
ANSWER: Thank you to the reviewer for reviewing our manuscript and for the suggestions made to improve our paper.
We modified Table 2 as per the suggestion, dividing the results into 3 columns and changed the title as per the suggestion.
In addition, we have added additional information to the table for completeness (such as follow up, MSTS score and complications).
Reviewer 2 Report
Comments and Suggestions for Authors
Comments to the authors
Thank you for inviting me to review the manuscript entitled “Use of custom-made prosthesis in upper limb oncological reconstruction: a systematic review”.
This is a systematic search of studies presenting the use of custom-made prostheses utilized in oncologic reconstruction of the upper limb.
Major concerns that need specific improvement:
The study lacks some fundamental part of the methodology of a systematic review, such as detailed indication of first and second screening stages of the study, quality assessment evaluation of the included studies, lack on details of reason for exclusion of the studies (flowchart fig 1). Moreover, all the information contained in the results section should also be provided in a table according to the data extracted. No indication about the satisfaction has been provided to respond to aim 2.
Some other minor points to be addressed are the following:
Abstract: the results presented in the abstract do not respond to the two aims of the systematic review, nor the conclusion is supported by the results section. For example, how did they test the reliability of the custom-made prosthesis? The satisfaction is not even mentioned. With the aim 1, are the authors testing the prevalence of use of custom-made prosthesis? Because if that is the case, they have to retrieve also those that did not utilize it. In other words, the manuscript may probably be based on review of complication
Introduction
Line 83: It is not clear what ‘level I-V evidence’ is. Is this the quality assessment of the article? This should not be assessed at the eligibility screening
How did the author assess the second aim of the study, if the satisfaction is not part of the data extraction (lines 97-101)?
Methods:
This section lacks completely important parts of the methodology of a systematic review.
Results: This should also include results of the quality assessment scale
Conclusions: not based on the current findings. It is difficult to assess the reliability with the current study design.
Reference: these are appropriate
Author Response
Comments to the authors
Thank you for inviting me to review the manuscript entitled “Use of custom-made prosthesis in upper limb oncological reconstruction: a systematic review”.
This is a systematic search of studies presenting the use of custom-made prostheses utilized in oncologic reconstruction of the upper limb.
Major concerns that need specific improvement:
The study lacks some fundamental part of the methodology of a systematic review, such as detailed indication of first and second screening stages of the study, quality assessment evaluation of the included studies, lack on details of reason for exclusion of the studies (flowchart fig 1). Moreover, all the information contained in the results section should also be provided in a table according to the data extracted. No indication about the satisfaction has been provided to respond to aim 2.
ANSWER: Thank you for the thorough and extensive review, which gave us the opportunity to revise the manuscript substantially and improve its quality. The methodology part was completely revised, including the flowchart part of the study selection, the evaluation of the studies according to Coleman's Score was added. The results section has also been implemented and all the results shown in table 2, which has also been revised. Finally, according to your suggestion, as we were not able to derive patient satisfaction from the included studies this second purpose of the study was eliminated, and the manuscript revised accordingly.
Some other minor points to be addressed are the following:
Abstract: the results presented in the abstract do not respond to the two aims of the systematic review, nor the conclusion is supported by the results section. For example, how did they test the reliability of the custom-made prosthesis? The satisfaction is not even mentioned. With the aim 1, are the authors testing the prevalence of use of custom-made prosthesis? Because if that is the case, they have to retrieve also those that did not utilize it. In other words, the manuscript may probably be based on review of complication
ANSWER: Thank you for your consideration. Indeed, it was not possible for us to calculate and evaluate the level of patient satisfaction, so this second aim of the study has been eliminated and discontinued. In addition, to better specify the purpose of our review to the reader, the title has also been changed, emphasising as you suggested that the analysis is focused on surgical outcomes and complications.
Introduction
Line 83: It is not clear what ‘level I-V evidence’ is. Is this the quality assessment of the article? This should not be assessed at the eligibility screening
ANSWER: Thanks for the suggestion, Level I-V refers to the quality of studies evaluated according to the American Academy of Orthopedic Surgeons (from which we have quoted). We put it in the inclusion criteria to highlight how articles were not excluded based on the level of evidence, but all were included.
How did the author assess the second aim of the study, if the satisfaction is not part of the data extraction (lines 97-101)?
ANSWER: thank you for the suggestion, effectively from the analysis of the studies it was not possible to calculate the level of patient satisfaction, so this was removed from the scope of the study. We apologise for the error.
Methods:
This section lacks completely important parts of the methodology of a systematic review.
ANSWER: Thank you for this important suggestion. The methodology part has been completely revised and divided into subsections, following the PRISMA guidelines. The flow-chart for the inclusion of studies was updated and the part on the evaluation of studies was added. To implement this, the PRISMA cheklist was compiled and added (available as supplementary material). Results: This should also include results of the quality assessment scale
Conclusions: not based on the current findings. It is difficult to assess the reliability with the current study design.
ANSWER: thanks for the suggestion. We have revised the conclusions by adapting it to the results of the study.
Reference: these are appropriate
Round 2
Reviewer 2 Report
Comments and Suggestions for Authors
The authors have now provided a revised version of their manuscript.
1) aim 2 in the introduction has not been deleted (i.e., satisfaction)
2) reviewes and other types of study should still be listed in the exclusion criteria. Normally systematic review does not include anything but randomized clinical trial. If case reports were included, a justification is most likely needed.
3) not clear why there are two flowcharts now. Just keep the PRISMA one, and delete the first one created
4) abbreviations explained in the footnote of the figures should be listed in alphabetic order
5) from the subsection created by the authors, it seems like all the results are under "demographics", which is not appropriate. Either the authors create other subsections, or they cannot include al these results under demographics
6) is there the possibility of completing table 2 in terms of sex difference also for the complications?
7) limitations of the reviews need to be included
Author Response
The authors have now provided a revised version of their manuscript.
1) aim 2 in the introduction has not been deleted (i.e., satisfaction)
ANSWER: thank you for the remark. We apologise for the transcription error, the second aim has been removed.
2) reviewes and other types of study should still be listed in the exclusion criteria. Normally systematic review does not include anything but randomized clinical trial. If case reports were included, a justification is most likely needed.
ANSWER: Thank you for your suggestion, which gives us the opportunity to further specify our selection of articles. We have amended section 2.2. on study selection to make it more precise.
The reviews were included in the search in order to subsequently analyse the bibliography to be able to identify additional studies that would otherwise have been lost, but no reviews were included among the papers in our review. On the other hand, case reports were included due to the small amount of literature and the few studies on the subject, in order to increase the sample size and to be able to have larger numbers.
3) not clear why there are two flowcharts now. Just keep the PRISMA one, and delete the first one created
ANSWER: In our manuscript file there is only one flowchart according to the PRISMA guidelines.
4) abbreviations explained in the footnote of the figures should be listed in alphabetic order
ANSWER: footnote has been revised according to your specifications.
5) from the subsection created by the authors, it seems like all the results are under "demographics", which is not appropriate. Either the authors create other subsections, or they cannot include al these results under demographics
ANSWER: thanks for the suggestion. We have modified the results section by dividing it into further subsections, so that the data is more organised.
6) is there the possibility of completing table 2 in terms of sex difference also for the complications?
ANSWER: Here we have not divided the results by sex because not all articles report this, so the data would be misleading and on small numbers.
7) limitations of the reviews need to be included
ANSWER: thanks for the suggestion. The limitations have been highlighted in a separate section (4.1) and modified by further specifying the scarce literature on the subject.